# Multivariate Analysis of Amino Acids and Health Beneficial Properties of Cantaloupe Varieties Grown in Six Locations in the United States

**DOI:** 10.3390/plants9091058

**Published:** 2020-08-19

**Authors:** Jashbir Singh, Rita Metrani, Guddadarangavvanahally K. Jayaprakasha, Kevin M. Crosby, Sadhana Ravishankar, Bhimanagouda S. Patil

**Affiliations:** 1Vegetable & Fruit Improvement Center, Department of Horticultural Sciences, Texas A&M University, 1500 Research Parkway, Suite A120, College Station, TX 77845-2119, USA; singh2014@tamu.edu (J.S.); ritametrani@tamu.edu (R.M.); k-crosby@tamu.edu (K.M.C.); 2National Center of Excellence for Melon at the Vegetable and Fruit Improvement Center, Texas A&M University, College Station, TX 77845, USA; 3School of Animal and Comparative Biomedical Sciences, University of Arizona, 1117 E. Lowell Street, Tucson, AZ 85721, USA; sadhravi@email.arizona.edu

**Keywords:** cantaloupe, amino acid, growing location, GABA, citrulline

## Abstract

Cantaloupe is a good dietary source of amino acids, including γ-aminobutyric acid (GABA), glutamine, and citrulline. However, the levels of these amino acids vary among different cantaloupe varieties grown in different locations. Understanding the variation in amino acid contents provides fundamentally important information for quality control and improving melon varieties. To examine this variation, we measured the amino acid contents in cantaloupes grown in six locations in the United States (Texas, Georgia, North Carolina, California, Indiana, and Arizona). Principal component analyses were applied to analyze the effect of growing location on the amino acid profiles in different varieties. The GABA content ranged from 1006.14 ± 64.77 to 3187.12 ± 64.96 µg/g and citrulline ranged from 92.65 ± 9.52 to 464.75 ± 34.97 µg/g depending on the variety and location. Total phenolic contents, α-amylase inhibition, and antioxidant activities were also measured. Tuscan type Da Vinci had significantly higher phenolic contents in Arizona (381.99 ± 16.21 µg/g) but had the lowest level when grown in California (224.56 ± 14.62 µg/g). Our analyses showed significant differences in amino acid levels, phenolics contents, and antioxidant activity in the cantaloupe varieties based on the growing location. These findings underline the importance of considering growing location in the selection and improvement of cantaloupe varieties.

## 1. Introduction

Melons (*Cucumis melo* L.) such as cantaloupe are important commercial crops and are consumed all over the world. Cantaloupe is favored by consumers due to its pleasant aroma, refreshing flavor qualities, nutrition, and phytochemical composition [1,2]. Melon fruits have high levels of the GABA [3] and other amino acids, particularly valine and alanine [4]. Amino acid identification and quantification of melon can be performed by reverse phase liquid chromatography, gas chromatography, and nuclear magnetic resonance [5,6,7]. Liquid chromatography is a commonly used technique for analysis of amino acids from plants. Analysis is mainly performed by liquid chromatography coupled with ultraviolet, diode-array, fluorescent, or mass detectors after derivatization. Commonly used derivatization reagents are dansyl-chloride, *O*-phthalaldehyde, 9-fluorenylmethyloxycarbonyl, and 6-aminoquinolyl-N-hydroxysuccinimidyl carbamate [5,8,9,10].

Amino acids play significant roles in plant growth and production of metabolic energy. They provide resistance to stress and help control cellular pH [11,12]. The amino acids citrulline and proline accumulate in different parts of the plant under stress conditions such as high light intensity or drought, and prevent oxidative injuries. Therefore, these amino acids can be used as biomarkers for selecting drought-tolerant cultivars [13,14]. Amino acids such as alanine, methionine, tryptophan, and phenylalanine are also involved in protein biosynthesis and the formation of secondary metabolites [15]. Branched and aromatic amino acids are precursors of volatile constituents; for example, phenylalanine is a precursor of alcohols and aldehydes, whereas isoleucine is a precursor of esters [16,17]. Amino acids such as proline, histidine, and asparagine also accumulate in heavy metal-stressed plants and play a significant role in metal binding [18].

In human nutrition, amino acids are important food compounds crucial for development, health, and reproduction. Amino acids act as a source of energy and precursors of proteins. They are involved in biological processes such as regulation of metabolism, reduction of blood sugar, and improvement of cardiovascular health, and they act as neurotransmitters and antioxidants [19,20,21]. Arginine is involved in nitric oxide production by NO synthase (NOS), where nitric oxide functions as signaling molecules; these potent vasodilators improve cardiovascular health, enhance sport performance, and lower the risks of stroke [22,23]. Certain amino acids function as excitatory (glutamate, aspartate) or inhibitory (GABA) neurotransmitters in the central and peripheral nervous systems. They have important roles in the pathogenesis of depression and mood disorders [24]. Amino acids also act as precursors for the gut microbiota to produce short chain fatty acids, which are anti-inflammatory [25,26]. For example, in the gut, anaerobic bacteria biosynthesize butyrate from threonine, lysine, and glutamate [27]. Glycine, ornithine, and glutamate are used for acetate biosynthesis and threonine is involved in the biosynthesis of acetate and propionate [28,29].

Free radical species have been reported to interfere with metabolic processes which lead to various chronic diseases [30,31]. Antioxidants such as phenolics play a crucial role in inhibiting oxidative stress and show a positive association with human health by strengthening immune defenses [32,33]. The antioxidant activity of plants can be determined by using different bioassays such as 2, 2-diphenyl-1-picrylhydrazyl (DPPH) and 2,2-azino-bis(3-ethyl-benzothiazoline-6-sulfonic acid) (ABTS) [34,35]. Melons have high phenolics content, including the major phenolic compounds hydroxybenzoic, hydroxycinnamic, and phenolic acid derivatives [36,37].

Different environmental parameters, such as temperature, light intensity, and growing locations, play a significant role in crop development and fruit maturation. The temperature causes the strongest impact on the fruit development as compared to other factors [38]. Melon crops are sensitive to air temperature and higher average temperature causes early fruit maturation [39]. Pardossi et al. reported melon plants grown in summer (24–25 °C) showed early maturation of fruit compared to the fruit grown in spring (20–22 °C) [39]. Similarly, light intensity was also a major limiting factor in melon development. Yang et al. found that low light intensity significantly influenced the sucrose and galactosyl-sucrose oligosaccharide metabolism in both melon fruits and leaves [40]. In melons, the levels of amino acids and antioxidants vary with growing conditions and genotypes [41,42,43,44]. Therefore, qualitative and quantitative analyses of fruits grown in different locations are essential for quality control and the production of improved varieties. Antioxidants such as phenolic compounds prevent the formation of free radical species or inhibit radical-chained reaction and protect human body cells from injuries [45]. Published studies showed that melon possesses therapeutic effects such as antioxidant, β-glucosidase, and α-amylase inhibitory activities [37,46]. However, assessment of the therapeutic effects of cantaloupe varieties grown in different locations is limited.

Therefore, we report the amino acid profiles, total phenolic contents, α-amylase, and antioxidant activities of different cantaloupe varieties harvested from six locations. To the best of our knowledge, this is the first report comparing the amino acid profiles and biological activities of cantaloupes grown in different states of the United States.

## 2. Results and Discussion

### 2.1. Quantitative Analysis of Amino Acids

We examined the influence of growing location on amino acid profiles, using different cantaloupe cultivars Western Shipper (F-39), Harper-type Infinite Gold (HT-IG), and Tuscan type Da Vinci (TT-DV) grown in six locations in the U.S.: Texas (TX), Georgia (GA), North Carolina (NC), California (CA), Indiana (IN), and Arizona (AZ). Along with these varieties, commercial local varieties (Primo (PRI), Athena (ATH), Alaniz gold (ALG), Caribbean king (CAR), and Cruiser (CRU)) were grown in TX, GA, NC, CA, IN, and AZ, respectively (Figure 1).

The combined free amino acid contents of cantaloupe varieties grown in different locations are presented in Figure 2. Combined free amino acids content in melon F-39 ranged from 5035.73 to 6318.7195 µg/g with the highest in IN and lowest in NC. In HT-IG, the amino acid content ranged from 5420.01 to 7682.76 µg/g and was higher in CA as compared with other locations. Similarly, in TT-DV melons, combined free amino acids ranged from 5011.22 to 8421.68 µg/g depending on the growing location. Among the commercial local varieties, CAR from CA had the highest (7086.10 µg/g) combined free amino acids followed by CRU (6907.68 µg/g) from IN but ATH fruits (4741.53 µg/g) from GA had the lowest value.

The comparative amino acids chromatograms of cantaloupe variety TT-DV grown in different locations are shown in Figure 3. Variation in the HPLC-FLD peak height of different amino acids in the TT-DV variety provides evidence of differences in amino acids content among the states. For example, the highest peak intensity for GABA (peak 12) was detected in NC and lowest in CA and GA. Appendix A depicts the HPLC-FLD chromatograms of amino acids from different cantaloupe varieties grown in Texas. The amino acid levels in the varieties and growing locations were compared for significance at α = 0.05 (Appendix A). Comparing fruits grown in the different locations, glutamine (Gln) was significantly higher in F-39 fruit harvested from TX (1227.04 ± 93.65 µg/g), followed by AZ (1092 ± 57.02 µg/g) compared to other locations (Figure 4). Compared also with other locations, TT-DV melons grown in IN had significantly higher levels of Gln (2034.94 ± 158.74 µg/g) followed by TX (1453.68 ± 57.25 µg/g) and NC (1406.29 ± 19.67 µg/g). For the commercial local varieties, Gln was significantly higher in fruit grown in TX (PRI) (1637.61 ± 60.04 µg/g) followed by IN (CRU) (1454.65 ± 58.61 µg/g) and NC (ATH) (1278.52 ± 120.43 µg/g) than in other locations (Figure 4). Our results are consistent with other studies that reported higher levels of Gln (2060.3 µg/g) in melon fruit compared to other amino acids [4].

In melon fruit, GABA levels ranged from 1006.14 ± 64.77 to 3187.12 ± 64.96 µg/g depending on variety and location. GABA content in F-39 was significantly higher in fruits grown in NC (1728.35 ± 56.01 µg/g) and IN (1850.58 ± 77.78 µg/g). Similarly, HT-IG fruits grown in NC showed significantly high GABA contents (3008.03 ± 128.01 µg/g) and TT-DV fruit from NC, IN, and AZ had high amounts of GABA—3187.12 ± 64.96, 2892.86 ± 121.98, and 2755.61 ± 216.86 µg/g, respectively (Figure 4).

Among the commercial local varieties (PRI, ATH, CAR, ALG, and CRU), there was no significant difference in GABA content. Another published study evaluated the GABA content in melon harvested in different seasons—autumn, winter, spring, and summer. They reported the highest content of GABA in melons harvested in summer (2010 µg/g) and the lowest (880 µg/g) in winter season, consistent with a strong effect of environment on the accumulation of GABA in melon fruits [3]. The Asp content was significantly higher in CA and AZ at 2388.45 ± 147.30 and 1500.88 ± 47.58 µg/g, respectively, than in other locations. Asp was higher in fruits grown in CA (1580.12 ± 116.98 µg/g) and AZ (1075.38 ± 107.70 µg/g) compared to other locations.

Levels of the nitric oxide precursors arginine (Arg) and citrulline (Cit) were significantly influenced by the growing conditions (Figure 5). Arg ranged from 38.17 ± 2.11 to 184.51 ± 18.18 µg/g and Cit ranged from 92.65 ± 9.52 to 464.75 ± 34.97 µg/g. Fruit F-39 grown in AZ had significantly higher contents of Arg (184.51 ± 18.18 µg/g) and Cit (240.28 ± 34.38 µg/g) followed by fruit grown in GA (140.75 ± 29.24 µg/g; 209.63 ± 44.03 µg/g) and TX (101.98 ± 7.15 µg/g; 206.74 ± 16.09 µg/g), respectively. Among the varieties, HT-IG fruit showed higher levels of Cit and HT-IG fruits grown in CA, TX, AZ, and IN showed no significant difference in Cit levels (464.75 ± 34.97, 457.25 ± 24.23, 436.05 ± 64.97, and 358.07 ± 58.38 µg/g, respectively; Figure 5). Similarly, TT-DV fruit had significantly higher Arg in AZ (123.85 ± 11.65 µg/g), while Cit was highest in IN (297.97 ± 21.33 µg/g) and TX (243.01 ± 10.50 µg/g) compared to other locations. Among the commercial local varieties, Arg content in ATH from NC (50.88 ± 5.99 µg/g) and GA (38.17 ± 2.11 µg/g) was the lowest and varieties CRU (CA) and ALG (AZ) had higher levels, 145.77 ± 12.5 and 114.25 ± 8.29 µg/g, respectively as compared to other varieties. Melon varieties CRU (IN) and ALG (AZ) had significantly higher levels of Cit at 360.52 ± 23.49 and 307.33 ± 58.84 µg/g, respectively, than the other varieties harvested from different locations. Citrulline, a non-essential amino acid and precursor to the arginine, is mainly present in Cucurbitaceae family members such as watermelon, squash, and cucumber [5]. A recent study reported the effect of two growing locations in NC on Cit and Arg levels, using different cultivars from those we studied here. They observed significant variation in the Cit and Arg levels within *Cucumis melo* cultivars. The Cit content ranged from 163 to 863 µg/g fresh weight and Arg ranged from 94 to 180 µg/g fresh weight [47]. These findings showed that amino acids contents are strongly influenced by growing locations and cultivars.

Melons are also rich in essential amino acids such as Val, Thr, Met, Trp, Phe, Ile, and Leu, which play a major role in fruit aroma, nutritional value, and health-promoting properties. Val content was high in all varieties grown in different locations and ranged from 243.75 ± 18.72 to 671.41 ± 4.50 µg/g (Table 1). Compared with the other locations, F-39 fruit grown in IN (671.41 ± 23.84 µg/g) had significantly high levels of Val, followed by NC (455.75 ± 3.54 µg/g) and TX (340.15 ± 14.09 µg/g). HT-IG showed no significant variation in Val content among the locations. TT-DV fruit had a significantly higher level of Val in AZ (541.50 ± 41.10 µg/g) and NC (509.08 ± 7.11 µg/g). According to the Food and Agriculture Organization (FAO) and the World Health Organization (WHO, 1985), daily requirements of amino acids are Trp (3.5 µg/g), Val (10 µg/g), Ile (10 µg/g), and Leu (14 µg/g) [48]. These show that melon is an excellent dietary source for essential amino acids.

Among the commercial local varieties, CRU grown in IN had higher Val content (510.73 ± 4.5 µg/g) and varieties harvested from TX (PRI), GA (ATH), CA (CAR) and AZ (ALG) had no significant difference. Thr content was significantly higher in F-39 fruit grown in AZ (132.49 ± 14.01µg/g) and GA (106.76 ± 21.28 µg/g), while in commercial local varieties, higher levels of Thr were measured in fruit grown in AZ (99.01 ± 5.55 µg/g) and CA (90.79 ± 11.25 µg/g). Similarly, HT-IG (100.28 ± 5.35 µg/g) and TT-DV (109.29 ± 11.08 µg/g) fruits showed higher Thr contents in AZ samples compared to other states. Trp is the precursor of serotonins, which regulate human mood, anxiety, appetite, and sleep [49]. Among the analyzed samples, HT-IG and the commercial local variety (CAR) harvested from CA showed significant higher contents of Trp at 71.82 ± 1.73 µg/g and 58.82 ± 2.04 µg/g, respectively, while TT-DV fruits grown in AZ had higher levels of Trp (68.67 ± 10.75 µg/g) compared to other locations and varieties.

The non-essential amino acid Ala was also identified in melon fruit, and was significantly higher in F-39 (494.13 ± 53.30 µg/g), HT-IG (477.01 ± 52.75 µg/g), and commercial local varieties (429.43 ± 46.75 µg/g) harvested from CA. Alanine was reported to be a major amino acid in melon and imparts sweetness to melon fruits [4]. Plants are exposed to various environmental stress factors such as drought, extreme temperature, chilling, and salinity. These factors significantly affect the numerous metabolites and metabolic pathways which influence plant growth and development [50,51]. Accumulation of amino acids under stress conditions has been reported in published literature [52,53,54]. Increases in the levels of specific amino acids have shown beneficial effect in a stressful environment. Accumulation of proline and citrulline has been correlated with plant stress tolerance. They act as a reactive oxygen species scavenger and provide protection to the plant under different environmental stresses [13,55]. Amino acids Asp and Glu are both involved in nitrogen remobilization in plants. Increases in the levels of Ser and Gly act as a biomarker for photorespiration [52,53,54,55,56]. Plants also accumulate GABA under various environmental stresses and this plays a vital role in osmotic and pH regulation, plant development, and carbon–nitrogen balance in plants [53,54,55,56,57]. A greenhouse study demonstrated that cold stress or low temperature (2 to 8 °C) exposure of spinach considerably increased the GABA content [58]. Similarly, in another study, cold exposure (−8 °C) induced significant increases in GABA content in wheat and barley [59]. In our study, higher levels of GABA were observed among the melons grown in the NC state (Figure 4). According to the weather data collected at the NC state, the minimum average temperatures in the months of March and April were −4.4 and 0 °C, respectively (Appendix A). Similarly, IN also had lower minimum temperatures, −5.5 °C for March and −2.7 °C for April, which might have been responsible for the higher level of GABA in melons harvested from these locations. In contrast, the average temperatures in AZ were 6.6 and 10 °C for the months of March and April, respectively, and low levels of GABA were observed in F-39 and HT-IG varieties. Based on these results, it can be inferred that a low temperature exposure during the early growth phase of melons may be an important contributing factor to enhance the GABA content in melon cultivars. There was also variation in the precipitation among the growing locations, but additional irrigation was applied in all locations, therefore, the amount of precipitation was probably not a contributing factor towards amino acids accumulation in melons.

These data suggest that no one variety-location combination produced the highest levels of amino acids, suggesting that breeding programs could produce cultivars with improved amino acid contents, but they must consider the growing locations and environmental stress factors. As observed in our data, GABA was high in all analyzed varieties grown in NC and IN locations, while F-39 and TT-DV grown in CA had low GABA content. The Gln content in TT-DV, HT-IG, and commercial local varieties was high in the IN location, while F-39 showed low content in the same location. Similarly, Cit was high in the IN location in all varieties except F-39. Moreover, another study reported that the variation in melon metabolites (volatiles, sugars, and amino acids) was affected by various factors such as the variety, growing season, planting dates, locations, and agricultural management practices [60].

### 2.2. PCA of Each Variety Grown in Different Locations

To further explore the interaction of variety and location, we conducted principal component analysis (PCA), which enables visualization of the location-amino acid contents space. Figure 6A–D shows the PCA of variation in amino acid profiles of each cantaloupe variety grown in different locations. The first (F1) and the second (F2) components explained 43.13% and 21.15% of the total variance, respectively. The effect of growing location on cantaloupe F-39 variety was distinguished based on the effect of Arg, Asn, Gln, Cit, Thr, Gly, Ala, GABA, Pro, and Val contents on component F1 and Ser, Asp, and Trp contents on component F2 (Figure 6A). Generally, the PCA indicated that melons grown in AZ had higher levels of Cit, Arg, Thr, and Asn, whereas melons grown in IN and NC had higher levels of Gly, Val, Pro, and GABA. Some differentiation among the states was also detected on component F2. CA-grown fruit had higher amounts of Asp, Ser, and Trp. Similarly, in variety HT-IG, states were differentiated based on the effect of Arg, Asn, Cit, Ser, Asp, Gly, Ala, Pro, Val, Trp, and Phe on component F1, and Hyp, Thr, β-Ala, GABA, Ile, and Leu on component F2 (Figure 6B). Overall, the PCA showed that HT-IG grown in GA had low levels of Arg, Asn, Cit, Gly, Ala, Pro, Trp, and Phe but HT-IG grown in CA had high levels of these amino acids. Likewise, HT-IG grown in NC had high amounts of Hyp, β- Ala, and GABA but HT-IG grown in TX had less of these amino acids. Overall, the first two principal components explained 71.92% of the total variation.

The growing locations for TT-DV were differentiated based on Ser, Thr, Gly, GABA, Pro, and Val on component F1 and Arg, Gln, Cit, Asp, Ala, Met, Trp, and Leu on component F2 (Figure 6C). Overall, examination of the location–amino acid content space indicated that the variety TT-DV grown in AZ and CA had higher levels of Arg and Asp. The first two principal components explained 66.05% of the total variation among the state average scores. Based on component F1, GA had lower levels of Ser, Gly, and Pro, while IN had higher levels of these amino acids. The commercial local varieties grown in different states were distinguished based on the effect of Arg, Asn, Cit, Ser, Thr, Gly, β-Ala, Ala, Pro, Val, Trp, Phe, and Ile on component F1 and Met and Leu on component F2 (Figure 6D). Examination of the location–amino acid level space indicated that the melons grown in IN had higher levels of Cit, Ser, Hyp, β-Ala, Val, and Ile. Melons grown in NC showed higher levels of Met and melons grown in CA had higher levels of Arg, Asn, Ala, and Trp. The first two principal components explained 64.63% of the total variation among the state average scores. The PCA results were supported by grouping of similar states in hierarchical cluster analysis (HCA) (Appendix A). States within each HCA cluster were also nearby in PCA, indicating that these states have similar amino acid profiles.

### 2.3. PCA of Different Varieties with a Single Growing Location

PCA also was conducted to visualize the cantaloupe variety–amino acid levels space for each growing location (Figure 7). To support PCA, HCA was performed to cluster cantaloupe varieties in each growing location (Appendix A).

The PCA and HCA showed that melon varieties were distinguished differently in each state. For example, melons grown in TX were differentiated based on the effect of Arg, Cit, Hyp, Gly, GABA, Val, Trp, Phe, lle, and Leu on component F1, and Asn, Ser, Asp, Thr, Ala, Pro, and Met on component F2 (Figure 7A). HT-IG and TT-DV differed from each other by amino acids that affected component F1. HT-IG had lower amounts of GABA, Leu, Val, and Hyp and TT-DV had higher levels of these amino acids. The first and second components jointly explained 79.04% of the total variance. Likewise, in GA-grown melon varieties, F-39 and ATH differed from each other based on the effects of Arg, Asn, Asp, Hyp, Thr, Pro, Met, and Val on component F1 (Figure 7B). F-39 melons had higher amounts of Arg, Cit, Gln, Ser, Pro, and Hyp and ATH melons had higher amounts of Asp, Gly, β-Ala, and Val. Based on component F2, TT-DV had higher amounts of Trp and GABA than other melon varieties. For the melon varieties grown in NC, the first and second components explained 81.98% of the total variance (Figure 7C). The melon varieties were differentiated based on the effects of Asn, Ser, Asp, Hyp, Gly, β-Ala, Pro, Val, and Leu on component F1, and Gln, Cit, Ala, Met, and Phe on component F2. The HT-IG variety had more amino acids affecting component F1, except Leu and Hyp, which were higher in variety ATH. F-39 had lower levels of amino acids (except Ala) driving component F2.

Melons grown in AZ were differentiated by PCA based on the effects of amino acids on component F1, except Arg, Thr, Met, lle, and Leu, which affected component F2 (Figure 7D). The first two components accounted for 88.88% of the total variance. TT-DV melons had higher amino acid contents driving component F1, except Asn, Gln, Cit, and β-Ala. This indicated that TT-DV was distinct from other melon varieties grown in AZ, with higher amounts of 10 amino acids. The first two components accounted for 88.04% of total variance in melons grown in CA (Figure 7E). Excluding Thr, GABA, Val, and Trp, melon varieties were differentiated based on the effect of the other amino acids on component F1. The first component accounted for 64.69% of the total variance and HT-IG differed from other varieties since it contained higher amounts of amino acids that were driving component F1, except Phe, lle, Leu, Asp, and Hyp, which were higher in TT-DV melons.

Similarly, the first two components accounted for 73.84% of the total variance in melon varieties grown in IN, based on the effects of Asn, Gln, Cit, Ser, Asp, Hyp, Ala, GABA, and Val on component F1, and Arg, Thr, Gly, β-Ala, Pro, lle, and Leu on component F2 (Figure 7F). TT-DV and F-39 differ from each other based on amino acids that affect component F1: if the amino acids that affect component F1 were high in TT-DV, then, they were low in F-39 and vice versa. Likewise, varieties APH and HT-IG were dissimilar to each other based on the amino acid contents driving component F2. The PCA results demonstrated that amino acids level in melon varieties grown in each location were significantly influenced by the cultivar. This suggested that the selection of cultivars that are suitable for the specific location is essential for achieving higher levels of amino acids.

### 2.4. Total Phenolics, α-Amylase and Antioxidant Activities

Cantaloupe varieties grown at different locations showed significant differences in total phenolics content and antioxidant activities (Figure 8A–D). The Folin–Ciocalteu (FC) method was used to estimate the total content of phenolics in cantaloupes. F-39 showed significantly higher total phenolics content in IN (489.76 ± 15.49 µg/g) followed by NC (419.72 ± 15.09 µg/g) and GA (387.10 ± 30.52 µg/g) compared to other locations (Figure 8A). There were no significant differences in HT-IG melons grown in all locations. TT-DV had higher total phenolic contents in NC (381.91 ± 33.82 µg/g) and AZ (381.99 ± 16.21 µg/g) and had the lowest level in CA (224.56 ± 14.62 µg/g). Among the commercial varieties, PRI had a significantly lower amount of phenolics (218.78 ± 13.09 µg/g) compared to other varieties. Antioxidant properties of phenolic compounds protect the body from free radical damage. Phenolic compounds exhibit several physiological properties such as antidiabetic, anti-inflammatory, antioxidant, cardio-protective, and antitumor activities [33,61].

High concentrations of phenolic compounds are reported in melons with the highest amounts in the rind, followed by seeds then pulp [37]. The present results are in agreement with previously reported total phenolics contents (243.8 µg/g) in cantaloupe [62], although another study reported lower total phenolics in cantaloupe fruit juice (95.35 ± 9.23 µg/g) and pulp (101.90 ± 14.99 µg/g) [63]. The variation in the total phenolics content may be due to cantaloupe cultivars and growing locations. The prominent phenolic compounds reported in melon have been isovanillic acid, coumaric acid-hexoside, hydroxyl-benzoic acid hexoside, chlorogenic acid, apigenin-7-α-glucoside, ferulic acid, and luteolin-7-O-glucoside [36,44]. The antidiabetic potential of cantaloupe was evaluated by determining their capacity to inhibit the α-amylase activity that is responsible for breakdown of polysaccharides to monosaccharides. The α-amylase activity of cantaloupe extracts of different varieties was presented in Figure 8B. Results demonstrated cantaloupe have hypoglycemic properties and there was no significant difference in α-amylase inhibition activity among the varieties and growing locations.

Previous studies have revealed that a strong correlation existed between phenolic compounds and inhibition of α-amylase enzyme. Phenolic compounds such as cinnamic acid, ferulic acid, chlorogenic acid, and caffeic acid were reported as potent inhibitors of α-amylase enzyme [64,65]. In a published study, polar extracts such as methanol and water had shown α-amylase inhibition, but their inhibition potential was lower than that of the hexane and chloroform extracts [66]. Similarly, inhibition of α-amylase activity of hexane and ethanol extracts of melon seeds was reported. It was found that although the ethanolic extract had a higher phenolics content as compared to the hexane extract, the α-amylase inhibiting activity was lower [46], suggesting that phenolic content is not the only factor determining α-amylase inhibiting activity and that other lipophilic or hydrophilic compounds may also play a role.

Free radical scavenging activity affects the synergistic or antagonistic behavior of phytonutrients, which define the nutritional and health-promoting functional qualities of a particular food. Reactive free radicals are associated with several chronic conditions such as inflammation, diabetes, cancer, and cardiovascular disease. Therefore, several recent studies supported the utilization of antioxidants as tools for disease management [32,67]. In the present study, extracts from cantaloupes from different growing locations and varieties were made and the free radical scavenging activity was determined by DPPH and ABTS assays. The DPPH assay is based on the reactivity of free radical and hydrogen donors. Antioxidants react with DPPH and form a 1,1,-diphenyl-2-picryl hydrazine adduct with a characteristic absorption at 517 nm [68]. DPPH free radical scavenging activity of F-39 from AZ (115.46 ± 10.60 µg/g) was significantly higher followed by IN (100.49 ± 2.65 µg/g) and CA (89.15 ± 7.72 µg/g) compared to other locations (Figure 8B). Similarly, TT-DV variety harvested from AZ (88.11 ± 2.85 µg/g) and TX (73.61 ± 8.77 µg/g) had significantly higher, while GA (51.85 ± 2.72 µg/g) had the lowest free radical scavenging activity. Among the commercial local varieties, higher DPPH activity was observed in ALG (91.10 ± 1.92 µg/g) as compared to varieties grown in different locations. Results of the ABTS assay showed F-39 harvested from TX (231.01 ± 56.85 µg/g) and IN (170.77 ± 7.56 µg/g) had significantly higher activity compared to other locations. Similarly, higher ABTS activity was found in TT-DV from TX (290.29 ± 53.57 µg/g), while in commercial local varieties, PRI from TX and ATH from GA had significantly higher activity 230.81 ± 46.16 and 169.73 ± 5.23 µg/g, respectively, than other varieties grown in different locations (Figure 8C,D).

There was variation in the activities obtained from DPPH and ABTS assays, possibly due to the type of reaction mechanisms, hydrogen atom transfer, and single electron transfer, respectively [68,69]. Moreover, the differences may reflect differences in the type of antioxidants present in the fruit, as DPPH˙ is scavenged by hydrophobic antioxidants and ABTS˙ is scavenged by hydrophilic and hydrophobic antioxidants. Our results are in agreement with previously reported ABTS free radical scavenging activity (153.3 ± 18.0 µg/g) in fresh-cut cantaloupe [62]. Similarly, in another study, ABTS activities reported for cantaloupe fruit juice and pulp were 323.21 ± 53.80 and 220.38 ± 40.50 µg/g, respectively [63]. Overall, higher antioxidant activities were reported in melon due to the presence of total phenolics [37,70]. Correlation between total phenolics and antioxidant activities were carried out with Pearson’s correlation coefficient (Appendix A). The total phenolics content of all melon varieties is positively correlated with antioxidant activity. Likewise, DPPH and ABTS are positively correlated in all varieties except commercial local varieties.

Antioxidant activity is also influenced by the structure of amino acids. Amino acids with strong reducing functional groups, such as hydroxy groups, attached with phenolic groups (tyrosine) and thiols (methionine and cysteine), have higher antioxidant activity than acidic (glutamic and aspartic) and neutral amino acids (proline, and serine) [71]. Several mechanisms, such as radical scavenging ability or chelation of metals which can quench singlet oxygen or breakdown hydroperoxide into free radicals, are proposed to explain antioxidant activity of amino acids [72,73]. For example, amino acids with thiol groups (methionine and cysteine) act as a metal chelator and represent 40%–80% of the total antioxidant activity of human serum albumin [74]. Methionine and cysteine also have the ability to reduce oxidative stress caused by lead [75]. Amino acids have also been shown to produce a synergistic effect with tocopherol which improves antioxidant activity [76]. Synergistic effects of methionine with epigallocatechin gallate (EGCG) and ascorbic acid have also been reported [77]. GABA can act as signaling molecules and exhibits reactive oxygen species (ROS) scavenging activity [78]. GABA activates the transcription of enzymes involved in phenolic biosynthesis, which promotes the accumulation of phenolic compounds and enhances antioxidant activity [79]. Yen et al. reported that GABA and total phenolic content were positively correlated during germination of hull-less barley. Moreover, phenolic, ferulic, and p-coumaric acids had a significantly positive correlation with GABA as compared to vanillic and caffeic acids [79]. Similarly, in tomato plants, GABA enhanced the level of gallic and coumaric acids under salinity conditions [80]. Our results show that variety F-39 had high GABA (Figure 4) and total phenolics content (Figure 8A) in NC and IN; this may be due to accumulation of phenolic acids such as ferulic and p-coumaric acids, which are known as prominent phenolic compounds in cantaloupe [36,44]. Similarly, the TT-DV variety also had high levels of GABA and total phenolics content in NC, IN, and AZ. Phenylalanine (aromatic amino acid) plays a significant role in plant metabolism through the phenylalanine ammonia-lyase (PAL) pathway and is involved in the formation and accumulation of phenolic compounds [81,82]. Our results show that Phe content in F-39 and TT-DV was higher in TX, IN, and AZ compared to other locations, which correlates with higher antioxidant activity of the fruits harvested from these locations. This may be due to the formation of phenolic compounds through the PAL pathway (Figure 8C,D).

The data were also analyzed in order to determine if the amino acid levels correlate with the antioxidant assays results (Figure 9, Figure 10, Appendix A). Amino acids Ser, Trp, and Met were positively correlated with antioxidant assays which vary with the variety. This suggested that amino acids play a significant role in antioxidant activities. Phongthai et al. reported that phenylalanine content in rice bran hydrolysate was positively correlated with DPPH and ABTS assays [83]. Figure 11 represents the tentative radical scavenging mechanism of selected amino acids. DPPH and ABTS assays usually measure the antioxidant behavior of the compounds through single electron transfer (SET) and also by hydrogen atom transfer (HAT) mechanism [84,85]. Aromatic amino acid, Trp, can act mainly through SET mechanism while hydroxy-containing amino acids such as Ser may act through HAT mechanism. Generally, both mechanisms occur in parallel, but depending on the amino acid structure and type of assays, one can dominate [84]. Ketnawa et al. reported that protein hydrolysates and peptides comprising a higher content of aromatic amino acids such as Tyr, Trp, and His have been linked with strong antioxidant activities via electron transfer mechanism [86].

Overall, similar to the results of the amino acids, the results of total phenolics and antioxidant activities showed significant variation among the varieties and growing locations, which may be considered as a factor for future melon breeding. Indeed, our results emphasize the need to consider location in breeding programs and suggest that there is ample room for improvement of the local adaptation of specific cultivars for growth in different regions. Therefore, our study will inform the selection of germplasm and location for future breeding efforts to improve the nutritional value of this popular, healthful fruit.

## 3. Material and Methods

### 3.1. Fruit Growing Locations

The different varieties of cantaloupe, Western Shipper (F-39), Harper type Infinite Gold (HT-IG), Tuscan type Da Vinci (TT-DV), and commercial local varieties Primo (PRI) from Texas (TX), Athena (ATH) from Georgia (GA) and North Carolina (NC), Alaniz gold (ALG) from Arizona (AZ), Caribbean king (CAR) from California (CA), and Cruiser (CRU) from Indiana (IN) were harvested from different growing locations, TX, GA, NC, CA, AZ, and IN, during the year 2018 (Figure 1). Weather data from all locations are given in Appendix A. Other growing conditions such as soil and irrigation are presented in Appendix A. Harvested fruits were shipped by carrier in refrigerated conditions to the Vegetable and Fruit Improvement Center (Texas A & M University, College Station, USA) for analysis.

### 3.2. Chemicals

Standard amino acids, L-aspartic acid (Asp), L-asparagine (Asn), L-serine (Ser), L-glutamine (Gln), L-citrulline (Cit), L-arginine (Arg), L-glycine (Gly), L-threonine (Thr), β-alanine (β-Ala), L-alanine (Ala), L-tryptophan (Trp), L-valine (Val), L-methionine (Met), L-phenylalanine (Phe), L-isoleucine (Ile), L-leucine (Leu), L-lysine (Lys), proline (Pro), hydroxyproline (Hyp), and γ-aminobutyric acid (GABA) were obtained from Sigma-Aldrich (St Louis, MO, USA). ACS and HPLC grade methanol, acetonitrile, formic acid, acetic acid, dansyl-chloride, di-amino-heptane, and triethyl amine (TEA) were obtained from Sigma-Aldrich (St. Louis, MO). Nano-pure water (resistivity 18.2 MΩ cm) was obtained from the NANO pure purification system (Barnstead/Thermolyne, Dubuque, IA). Sodium borate was purchased from Thermo Fisher Scientific (Waltham, MA, USA).

### 3.3. Sample Preparation

Cantaloupes were cut horizontally into two equal halves. The pulp was removed from the rind and cut into small pieces. Samples were blended for 1 min at maximum speed (Oster Blender with 12 speeds, 450 W). The blended samples (15 g) were mixed with 10 mL of methanol. The contents were homogenized for 2 min at 10,000 rpm (850 Homogenizer, Thermo Fisher Scientific, Waltham, MA, USA) and sonicated for 30 min. The extracts were centrifuged at 7826× *g* for 10 min and supernatants were filtered through Whatman filter paper No. 1. The residues were re-extracted twice with methanol (2 × 7 mL) to ensure complete extraction. The resultant supernatants were pooled and the final volume was recorded.

### 3.4. Amino Acid Derivatization

A 350-µL sample of the extracted supernatant was used for dansyl-chloride (DNS-CL) derivatization. To the above aliquot of the supernatant, 125 µL DNS-CL (0.01% in acetone), 415 µL of 5 mM sodium borate buffer, and 50 µL diamino heptane (internal standard 500 ppm) were added. Reaction mixtures were vortexed and incubated for 30 min in a water bath (60 ± 1 °C). Derivatization was stopped by adding 2 N acetic acid (50 µL). Later, the reaction mixture was centrifuged at 10,621× *g* for 5 min and the supernatant was analyzed by HPLC-FLD.

### 3.5. HPLC-FLD for Quantitation of Amino Acids

The chromatographic separation of dansyl derivatives of amino acids was performed on an HPLC system comprising a PerkinElmer Series 200 binary pump and an autosampler (Shelton, CT, USA). A 1260 Infinity fluorescence detector (Agilent Technologies, Santa Clara, CA, USA) at excitation (293 nm) and emission (492 nm) wavelengths were used to observe the peak response. Amino acids were separated on an Eclipse XDB-C8 column (4.6 × 150 mm, 5 μm) using binary gradient mobile phase—solvent A, 1% formic acid in water and solvent B, acetonitrile: formic acid: TEA (98:1:1, *v*/*v*). The gradient program was set with a flow rate of 0.6 mL min−1: isocratic 15% B (4 min), 15% to 20% B (8 min), 20% to 45% B (2 min) and kept isocratic 45% B (2 min), 45% to 50% B (2 min), 50% to 100% B (2 min), kept isocratic 100% B (5 min), 100% to 15% B (2 min), and remained isocratic for 2 min. The column was equilibrated for 4 min before the next injection. Five μL of derivatized sample was injected and column temperature was set at 30 °C.

### 3.6. Total Phenolic Content, α-Amylase, and Antioxidant Activities

#### 3.6.1. Total Phenolic Content

The Folin–Ciocalteu (FC) reagent was used for the determination of total phenolic content from melon samples as described in our previous publication [87]. Briefly, extracted samples (10 μL) were added to different wells of a microplate and adjusted to 100 μL with methanol. An amount of 40 μL FC reagent (25%) was added and incubated for 10 min, then sodium carbonate (50 μL) was added to each well and incubated for further 20 min. The absorbance at 760 nm was read using a KC-4 Microplate Reader (BioTek Instruments, Winooski, VT, U.S.A.). A standard curve was developed using various concentration of gallic acid (0, 10, 20, 30, 40, 50, 75, and 100 μg/mL). Results were expressed as gallic acid equivalents.

#### 3.6.2. α-Amylase Activity

Inhibition of α-amylase was performed according to published protocol [66]. Aliquots (10 μL) of melon extract were pipetted into a 96-well micro plate and final volume was adjusted to 140 μL with a 1% saline solution. To each well, 45 μL of starch (1%) were added followed by 45 μL of amylase (1 unit/mL) to initiate the reaction. The reaction mixture was incubated at 25 °C for 60 min. Fifty μL of 3,5-dinitrosalicylic acid was added to the sample and standard wells followed by incubation for 60 min at 50 °C. Change in color was measured at 540 nm. Acarbose (80 μg/mL) and methanol were used as positive and negative controls, respectively. Different concentrations of dextrose (equivalent to 25, 50, 75, 100, 125, and 150 μg) were prepared to develop a standard curve.

#### 3.6.3. 2,2-Diphenyl-1-Picrylhydrazyl (DPPH) Assay

DPPH˙ radical scavenging activity was measured according to our published protocols [88]. Briefly, 10 μL aliquots of methanol extract were added to 96-well micro plates (KC-4 Microplate Reader) and adjusted to 100 μL with methanol. An aliquot of 180 μL of methanolic DPPH solution (0.1 mM) was added to each well. A reaction mixture was incubated for 20 min in the dark and the absorbance was recorded at 515 nm. Standard ascorbic acid was used to prepare the calibration curve and results were expressed as µg/g ascorbic acid equivalent.

#### 3.6.4. 2,2′-Azino-Bis (3-ethylbenzothiazoline-6-sulfonic acid) (ABTS·) Assay

ABTS· assay was performed according to the methodology reported in our published method [68]. Briefly, ABTS˙ was prepared by mixing 7 mmol/L aqueous ABTS solution and 2.45 mmol/L potassium persulfate (K2S2O8) solution. The resultant solution was kept in the dark for 16 h. Fresh standard ascorbic acid and methanol extract were prepared, aliquots (10 μL) were pipetted into 96-well micro plates, and the total volume of each well was adjusted to 40 μL with methanol. Freshly prepared ABTS solution (initial absorbance was 0.7 at 734 nm) was added to each well and absorbance was measured at 734 nm using a KC-4 Microplate Reader (BioTek Instruments, Inc., Winooski, VT, USA).

### 3.7. Statistical Analysis

Data were analyzed by principal component analysis (PCA) using Pearson’s correlation method, Hierarchical Cluster analysis, and one-way analysis of variance (ANOVA) using XLSTAT software (Addinsoft, Paris, France). Five cantaloupe samples were analyzed from each variety and duplicate measurements (n = 5 × 2) were performed for each sample. Significant differences between means were observed by Tukey’s HSD test at a 5% probability level (*p* ≤ 0.05).

## 4. Conclusions

In the present study, we performed comparative profiling of amino acids, phenolic contents α-amylase, and antioxidant activities in melons harvested from six different growing locations (Texas, Georgia, North Carolina, California, Indiana, and Arizona). Statistical analyses showed significant differences in amino acid levels, phenolics contents, and antioxidant activity in the melon varieties based on the growing location. Cantaloupes are good sources of amino acids, especially the neurotransmitters, nitric oxide precursors, and essential amino acids. PCA differentiated the melon varieties in all states by amino acid content. In addition, PCA results were supported by HCA grouping of the states based on the similarity in amino acid content. The findings from this study provide important insight into the influence of growing location on the accumulation of amino acids and phenolic compounds among the different cantaloupe varieties and could prove valuable for melon improvement.

## Figures and Tables

**Figure 1 plants-09-01058-f001:**
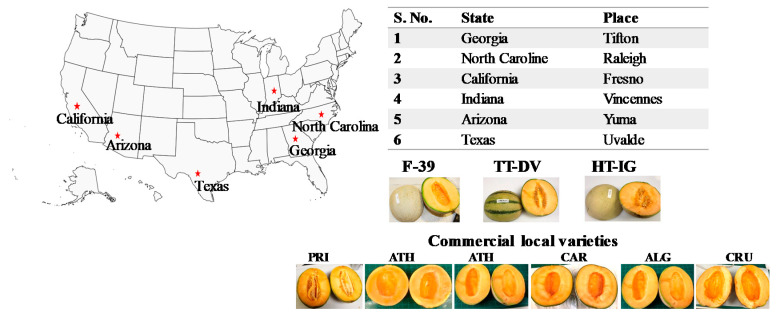
Cantaloupe varieties grown in different locations in the U.S. Cantaloupe varieties: Western Shipper (F-39), Tuscan type Da Vinci (TT-DV), Harper-type Infinite Gold (HT-IG); commercial local varieties: Primo (PRI) from Texas, Athena (ATH) from Georgia and North Carolina, Caribbean king (CAR) from California, Alaniz gold (ALG) from Arizona, and Cruiser (CRU) from Indiana.

**Figure 2 plants-09-01058-f002:**
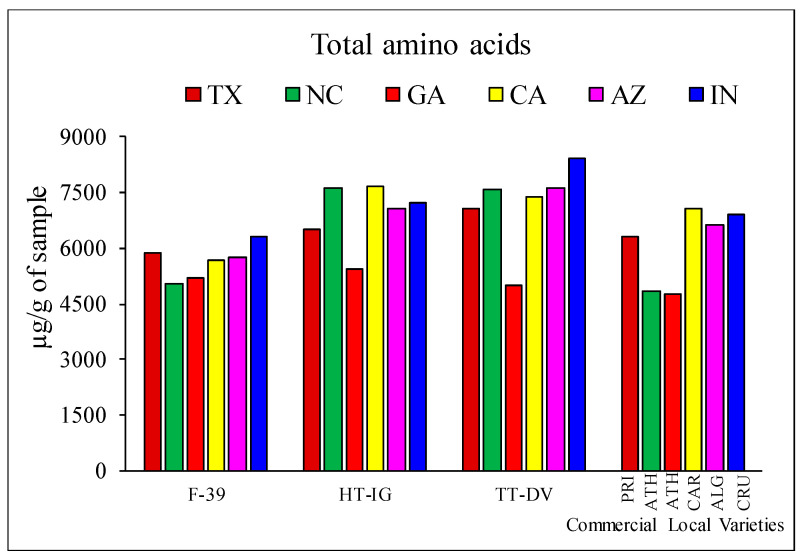
Combined free amino acid contents in cantaloupe varieties grown in different locations: Texas (TX), Georgia (GA), North Carolina (NC), California (CA), Indiana (IN), and Arizona (AZ). Cantaloupe cultivars: Western Shipper (F-39), Harper-type Infinite Gold (HT-IG), and Tuscan type Da Vinci (TT-DV); commercial local varieties: Primo (PRI) from TX, Athena (ATH) from GA and NC, Alaniz gold (ALG) from AZ, Caribbean king (CAR) from CA, and Cruiser (CRU) from IN.

**Figure 3 plants-09-01058-f003:**
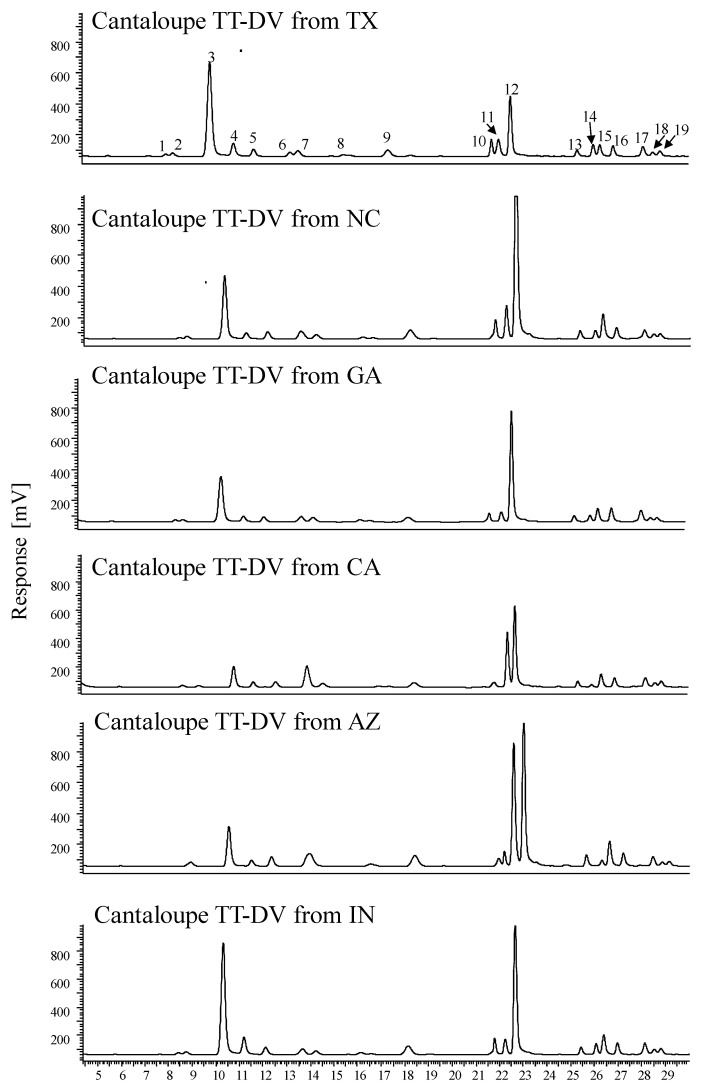
Comparative HPLC-FLD chromatograms of amino acids from cantaloupe variety—Tuscan type Da Vinci (TT-DV) grown in Texas (TX), Georgia (GA), North Carolina (NC), California (CA), Indiana (IN), and Arizona (AZ). Amino acids: (1) arginine; (2) asparagine; (3) glutamine; (4) citrulline; (5) serine; (6) aspartic acid; (7) hydroxy proline; (8) threonine; (9) glycine; (10) β-alanine; (11) alanine; (12) γ-aminobutyric acid; (13) proline; (14) methionine; (15) valine; (16) tryptophan; (17) phenylalanine; (18) isoleucine; (19) leucine.

**Figure 4 plants-09-01058-f004:**
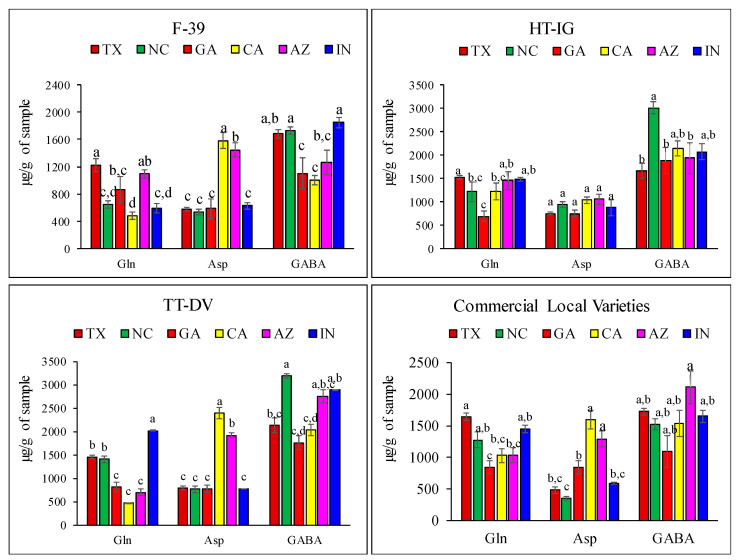
Levels of amino acids (Gln, Asp, and GABA) in different varieties of cantaloupe harvested from different locations: Texas (TX), Georgia (GA), North Carolina (NC), California (CA), Indiana (IN), and Arizona (AZ). Cantaloupe varieties: Western Shipper (F-39), Harper-type Infinite Gold (HT-IG), and Tuscan type Da Vinci (TT-DV); commercial local varieties: Primo from TX, Athena from GA and NC, Alaniz gold from AZ, Caribbean king from CA, and Cruiser from IN. Means with the same letter indicate no significant difference (*p*  <  0.05). Data represent means ± SE of n = 10. Gln—glutamine; Asp—aspartic acid; GABA—γ-aminobutyric acid.

**Figure 5 plants-09-01058-f005:**
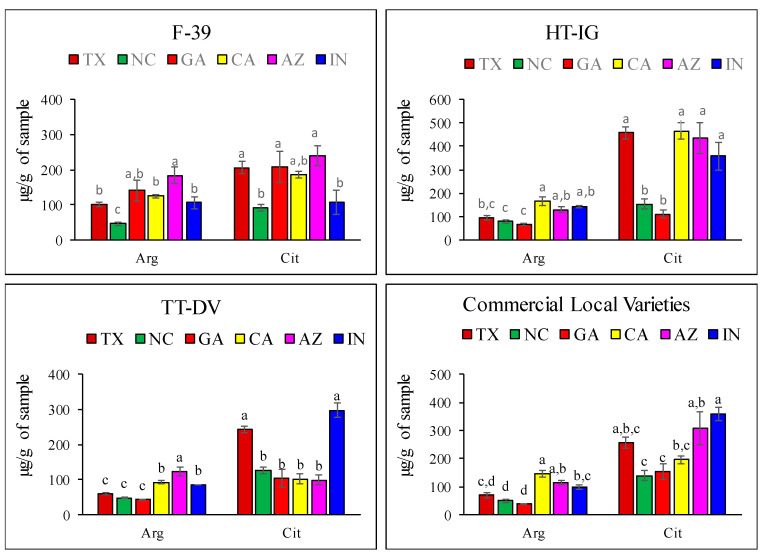
Levels of nitric oxide precursors in cantaloupe varieties harvested from different locations: Texas (TX), Georgia (GA), North Carolina (NC), California (CA), Indiana (IN), and Arizona (AZ). Cantaloupe varieties: Western Shipper (F-39), Harper-type Infinite Gold (HT-IG), and Tuscan type Da Vinci (TT-DV); commercial local varieties: Primo from TX, Athena from GA and NC, Alaniz gold from AZ, Caribbean king from CA, and Cruiser from IN. Means with the same letter indicate no significant difference (*p*  <  0.05). Data represent means ± SE of n = 10. Arg—arginine; Cit—citrulline.

**Figure 6 plants-09-01058-f006:**
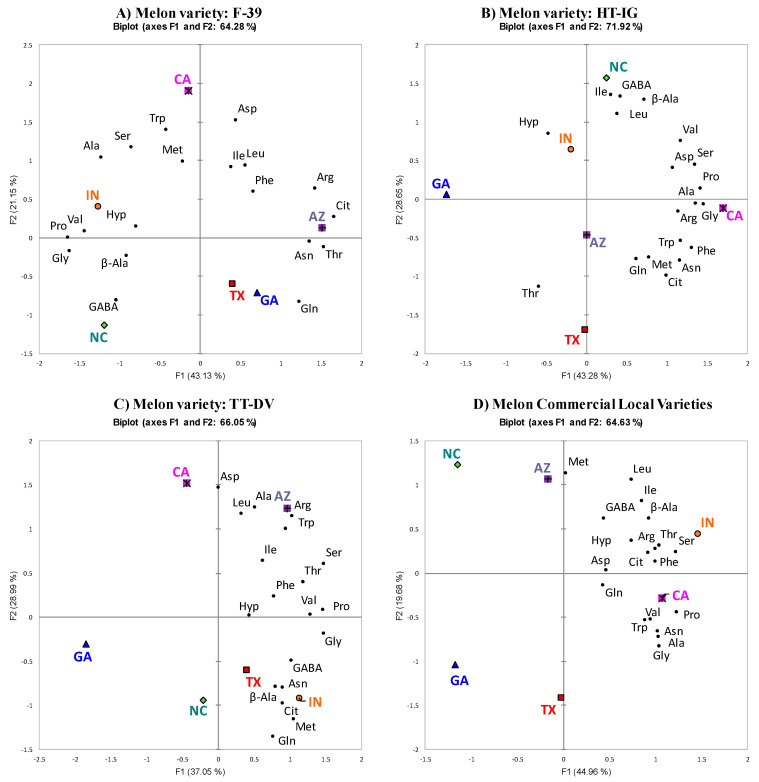
(**A**–**D**). Principal component analysis of average amino acids scores for each variety grown at different locations. Growing locations: Texas (TX), Georgia (GA), North Carolina (NC), California (CA), Indiana (IN), and Arizona (AZ). Cantaloupe varieties: Western Shipper (F-39), Harper-type Infinite Gold (HT-IG), and Tuscan type Da Vinci (TT-DV). Commercial local varieties: Primo from TX, Athena from GA and NC, Alaniz gold from AZ, Caribbean king from CA, and Cruiser from IN. Abbreviations: Asn—asparagine; Ser—serine; Hyp—hydroxy proline; Thr—Threonine; Gly—glycine; β-Ala—beta-alanine; Ala—alanine; Pro—proline; Met—methionine; Val—valine; Trp—tryptophan; Phe—phenylalanine; Ile—isoleucine; Leu—leucine; Arg—arginine; Cit—citrulline Gln—glutamine; Asp—aspartic acid and GABA—γ-aminobutyric acid.

**Figure 7 plants-09-01058-f007:**
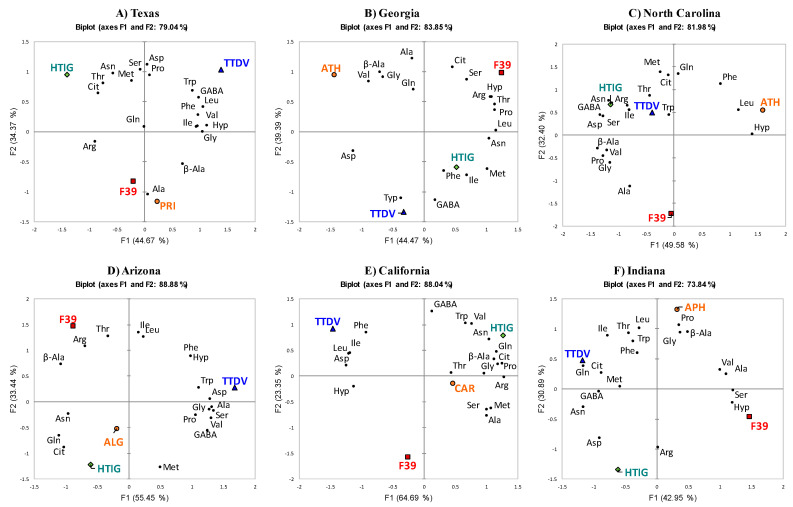
(**A**–**F**). Principal component analysis of average amino acid scores of different melon varieties grown in Texas (TX), Georgia (GA), North Carolina (NC), Arizona (AZ), California (CA), and Indiana (IN). Cantaloupe varieties: Western Shipper (F-39), Harper-type Infinite Gold (HT-IG), and Tuscan type Da Vinci (TT-DV). Commercial local varieties: Primo (PRI) from TX, Athena (ATH) from GA and NC, Alaniz gold (ALG) from AZ, Caribbean king (CAR) from CA, and Cruiser (CRU) from IN. Asn—asparagine; Ser—serine; Hyp—hydroxy proline; Thr—Threonine; Gly—glycine; β-Ala—beta-alanine; Ala—alanine; Pro—proline; Met—methionine; Val—valine; Trp—tryptophan; Phe—phenylalanine; Ile—isoleucine; Leu—leucine; Arg—arginine; Cit—citrulline; Gln—glutamine; Asp—aspartic acid; GABA—γ-aminobutyric acid.

**Figure 8 plants-09-01058-f008:**
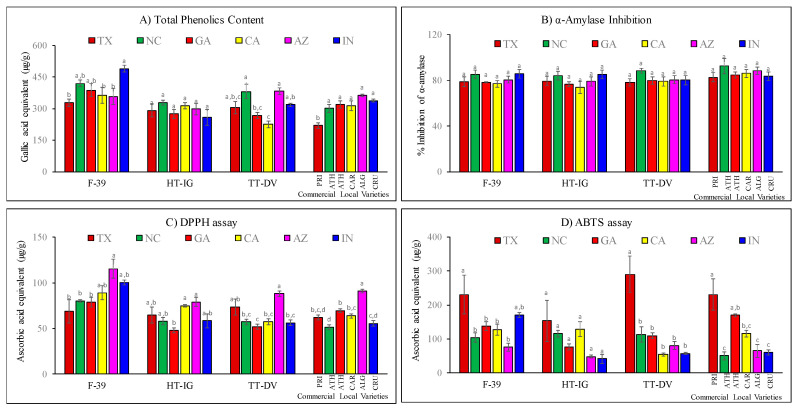
(**A**–**D**). Comparison of (**A**) total phenolic contents, (**B**) α-amylase inhibition (**C**) DPPH assay and (**D**) ABTS assay results of cantaloupe varieties grown in different locations—Texas (TX), Georgia (GA), North Carolina (NC), California (CA), Indiana (IN), and Arizona (AZ). Cantaloupe varieties: Western Shipper (F-39), Harper-type Infinite Gold (HT-IG), and Tuscan type Da Vinci (TT-DV). Commercial local varieties: Primo (PRI) from TX, Athena (ATH) from GA and NC, Alaniz gold (ALG) from AZ, Caribbean king (CAR) from CA, and Cruiser (CRU) from IN. Data represent means ± SE of n = 10. Means with the same letter indicate no significant difference (*p*  <  0.05).

**Figure 9 plants-09-01058-f009:**
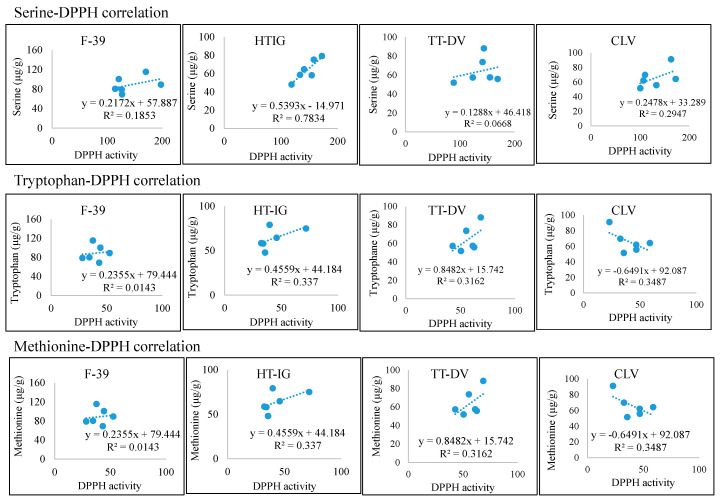
The relationship between selected amino acids and DPPH assay. Cantaloupe varieties: Western Shipper (F-39), Harper-type Infinite Gold (HT-IG), and Tuscan type Da Vinci (TT-DV). Commercial local varieties (CLV): Primo (PRI) from TX, Athena (ATH) from GA and NC, Alaniz gold (ALG) from AZ, Caribbean king (CAR) from CA, and Cruiser (CRU) from IN.

**Figure 10 plants-09-01058-f010:**
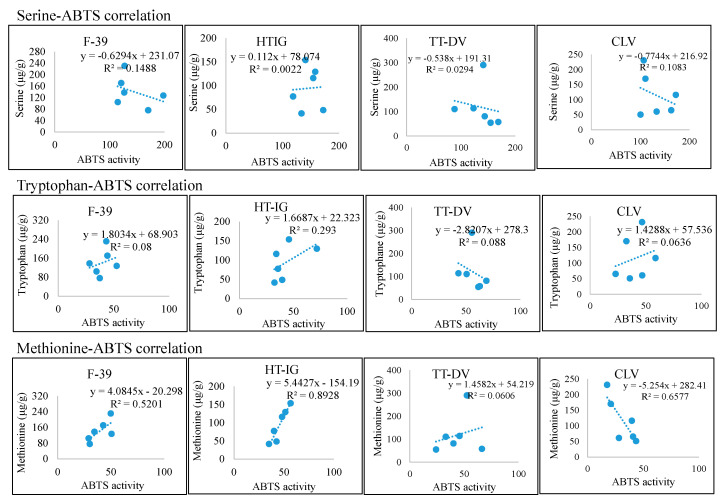
The relationship between selected amino acids and ABTS assay. Cantaloupe varieties: Western Shipper (F-39), Harper-type Infinite Gold (HT-IG), and Tuscan type Da Vinci (TT-DV). Commercial local varieties (CLV): Primo (PRI) from TX, Athena (ATH) from GA and NC, Alaniz gold (ALG) from AZ, Caribbean king (CAR) from CA, and Cruiser (CRU) from IN.

**Figure 11 plants-09-01058-f011:**
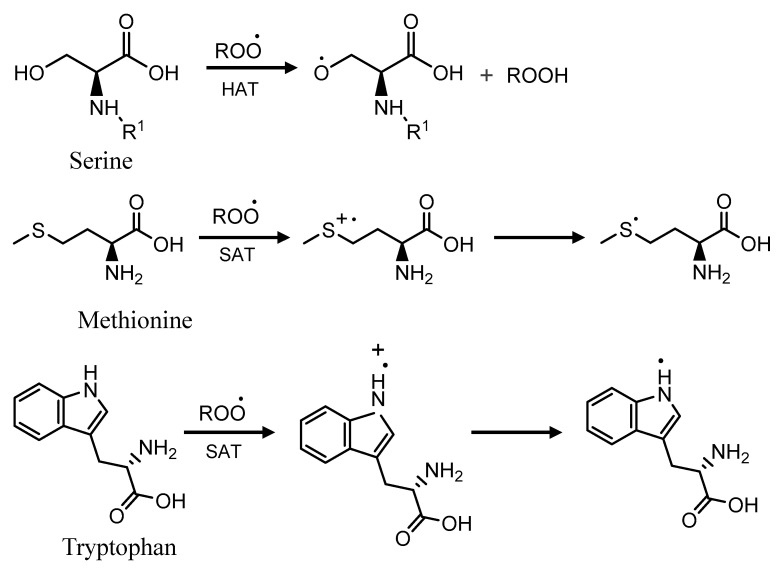
Radical scavenging mechanism of amino acids. HAT—hydrogen atom transfer; SET—single electron transfer.

**Table 1 plants-09-01058-t001:** Amino acid contents in cantaloupe varieties grown in different locations: Texas (TX), North Carolina (NC), Georgia (GA), California (CA), Arizona (AZ), and Indiana (IN). Data represent means ± SE of n = 10.

A. Amino acid contents in F-39 fruit grown in different locations.						
**Amino Acid**	**TX**	**NC**	**GA**	**CA**	**AZ**	**IN**
Asn	467.24 ± 30.95a	164.34 ± 31.28c	343.22 ± 69.86ab	278.4 ± 57.9bc	414.8 ± 52.37ab	270.37 ± 36.04bc
Ser	127.06 ± 8.39b	114.52 ± 16.43b	126.35 ± 23.64b	170.07 ± 12.84ab	120.86 ± 18.22b	197.57 ± 4.78a
Hyp	374.62 ± 45.16c	467.8 ± 41.09c	783.54 ± 151.41b	521.89 ± 44.67c	357.72 ± 30.26c	960.82 ± 34.29a
Thr	53.99 ± 12.67b	10.48 ± 4.8c	106.76 ± 21.28a	36.02 ± 3.03bc	132.49 ± 14.01a	39.38 ± 2.82bc
Gly	80.73 ± 6.47ab	92.05 ± 2.41a	62.45 ± 13.2bc	74.8 ± 8.42abc	53.58 ± 7.36c	95.06 ± 4.82a
β-Ala	34.59 ± 4.32b	74.59 ± 2.77a	35.73 ± 6.71b	39.54 ± 3.8bc	63.86 ± 6.64a	78.49 ± 4.31a
Ala	253.12 ± 54.99bc	355.59 ± 28.34abc	272.67 ± 63.77bc	494.13 ± 53.3a	178.64 ± 38.06c	405.26 ± 38.52ab
Pro	78.42 ± 6.17abc	99.02 ± 5.73a	68.13 ± 13.57bc	85.65 ± 11.66abc	61.13 ± 7.65c	90.59 ± 0.92ab
Met	49.84 ± 3.06a	29.14 ± 1.85b	34.6 ± 6.91ab	50.66 ± 8.85a	30.21 ± 4.07b	42.78 ± 1.73ab
Val	340.15 ± 14.09c	455.75 ± 3.54b	324.42 ± 69.29c	357.7 ± 33.51bc	287.86 ± 30.32c	671.41 ± 23.84a
Trp	43.3 ± 5.2a	34.52 ± 7.33ab	28.26 ± 4.49b	52.64 ± 5.59ab	37.54 ± 4.62ab	44.31 ± 1.17a
Phe	111.02 ± 14.66a	40.83 ± 2.93c	66.1 ± 13.6bc	82.65 ± 11.36ab	87.83 ± 6.81ab	88.64 ± 4.51ab
Ile	20.11 ± 1.79b	21.35 ± 0.63ab	19.13 ± 3.8b	23.13 ± 2.08ab	25.46 ± 0.78a	22.23 ± 0.59ab
Leu	19.85 ± 1.28c	25.78 ± 1.79bc	28.01 ± 5.42b	31.78 ± 1.96ab	34.8 ± 1.28a	27.98 ± 0.98bc
B. Amino acid contents in HT-IG fruit grown in different locations.						
**Amino Acid**	**TX**	**NC**	**GA**	**CA**	**AZ**	**IN**
Asn	572.79 ± 22.58a	468.04 ± 20.24ab	395.28 ± 23.86b	553.82 ± 11.97a	493.44 ± 75.74ab	505.65 ± 14.18ab
Ser	140.68 ± 3.02ab	154.28 ± 5.55ab	118.71 ± 4.02b	172.17 ± 5.99a	133.58 ± 20.71ab	157.97 ± 15.53ab
Hyp	288.94 ± 31.88c	414.74 ± 46.25bc	543.95 ± 65.91ab	436.05 ± 21.48bc	282.21 ± 40.8c	628.38 ± 17.4a
Thr	81.19 ± 30.44ab	32 ± 2.4b	72.59 ± 16.47ab	45.59 ± 2.41ab	100.28 ± 5.35a	52.92 ± 2.18ab
Gly	76.88 ± 10.4ab	80.64 ± 5.66ab	47.7 ± 4.73b	89.97 ± 2.72a	71.38 ± 12.13ab	63.53 ± 6.05ab
β-Ala	20.41 ± 5.42c	81.39 ± 7.5a	33.14 ± 2.88bc	66.56 ± 4.41a	55.03 ± 9.36ab	74.49 ± 10.28a
Ala	206.13 ± 4.98b	231.39 ± 25.96b	158.38 ± 18.4b	477.01 ± 52.75a	248.04 ± 61b	196.59 ± 34.59b
Pro	99.2 ± 7.53ab	110.5 ± 13.11a	65.04 ± 6.13b	122.27 ± 4.19a	83.63 ± 11.75ab	84.52 ± 10.38ab
Met	56.67 ± 0.99a	48.17 ± 4.4a	40.11 ± 5.26a	51.48 ± 3.77a	42.59 ± 7.1a	35.04 ± 6.08a
Val	345.22 ± 9.67a	445.42 ± 22.07a	318.36 ± 25.52a	424.48 ± 22.62a	383.35 ± 67.48a	350.47 ± 39.31a
Trp	45.84 ± 0.78b	34.07 ± 2.8b	35.63 ± 2.44b	71.82 ± 1.73a	39.61 ± 6.99b	32.34 ± 1.7b
Phe	68.85 ± 7.73ab	53.61 ± 5.64b	50.36 ± 6.17b	88.96 ± 4.54a	66.77 ± 7.5ab	62.48 ± 1.26b
Ile	15.58 ± 0.92a	39.52 ± 20.28a	17.83 ± 0.33a	21.09 ± 1.83a	19.87 ± 2.2a	25.82 ± 0.29a
Leu	18.63 ± 1.66b	27.35 ± 1.84ab	23.43 ± 3.05ab	27.12 ± 2.97ab	27.63 ± 3.29ab	31 ± 0.65a
C. Amino acid contents in TT-DV fruit grown in different locations.						
**Amino Acid**	**TX**	**NC**	**GA**	**CA**	**AZ**	**IN**
Asn	509.47 ± 13.85a	336.83 ± 19.67b	244.81 ± 19.47b	331.7 ± 18.49b	267.46 ± 47.15b	482.57 ± 7.08a
Ser	141.19 ± 5.92ab	123.19 ± 6.39ab	88.39 ± 5.65c	143.7 ± 6.13ab	168.77 ± 10.23a	154.69 ± 1.39a
Hyp	537.19 ± 254.53a	446.47 ± 25.48a	409.15 ± 27.68a	565.07 ± 18.86a	401.96 ± 23.01a	531.02 ± 18.26a
Thr	60.6 ± 6.93ab	64.2 ± 33.45ab	29.57 ± 2.61b	41.03 ± 3.71ab	109.29 ± 11.08a	59.22 ± 1.34ab
Gly	99.96 ± 8.47a	91 ± 3.98ab	56.12 ± 7.06c	68.89 ± 1.9bc	108.16 ± 9.78a	97.31 ± 1.69a
β-Ala	30.33 ± 5.69b	72.08 ± 4.65a	30.74 ± 3.68b	32.34 ± 2.88b	47.37 ± 2.41b	75.52 ± 3.05a
Ala	201.64 ± 32.17bc	149.02 ± 10.53bcd	98.16 ± 47.04cd	252.98 ± 34.31b	443.81 ± 57.3a	50.11 ± 10.05d
Pro	99.63 ± 6.66a	78.37 ± 3.15abc	56.65 ± 3.71c	72.93 ± 5.81bc	102.81 ± 10.77a	89.82 ± 3.78ab
Met	52.48 ± 2.77b	45.68 ± 2.45b	33.01 ± 2.39cd	24.03 ± 2.66d	40.01 ± 4.91bc	66.28 ± 1.25a
Val	409.28 ± 20.74b	509.08 ± 27.11a	303.99 ± 24.47c	388.01 ± 11.56bc	541.5 ± 41.1a	475.65 ± 6.22ab
Trp	55.38 ± 3.14ab	42.92 ± 4.2b	50.52 ± 4.11ab	61.27 ± 5.82ab	68.67 ± 10.75a	62.67 ± 1.66ab
Phe	145.48 ± 11.3a	59.44 ± 8.7c	81.61 ± 8.82bc	111.88 ± 7.03ab	104.82 ± 17.52ab	117.68 ± 1.87ab
Ile	22 ± 1.16c	20.17 ± 1.35c	20.31 ± 1.15c	30.55 ± 1.66a	23.53 ± 2.11bc	29.67 ± 0.68ab
Leu	24.71 ± 1.41c	25.4 ± 1.44bc	23.8 ± 1.91c	47.01 ± 2.57a	32.97 ± 3.63bc	33.78 ± 1.23b
D. Amino acid contents in commercial local varieties grown in different locations.						
**Amino Acid**	**TX**	**NC**	**GA**	**CA**	**AZ**	**IN**
Asn	454.28 ± 17.55a	147.21 ± 24.61d	206.74 ± 30.48cd	464.6 ± 21.43a	277.92 ± 30.72bc	361.34 ± 24.49ab
Ser	106.71 ± 5.77c	100.28 ± 10.76c	110.13 ± 7.98c	162.78 ± 8ab	133.04 ± 12.3bc	172.27 ± 4.18a
Hyp	420.85 ± 19.55bcd	557.59 ± 37.21ab	406.19 ± 23.23cd	533.1 ± 50.55bc	343.34 ± 42d	692.04 ± 13.03a
Thr	55.6 ± 9.35bc	23.66 ± 4.12c	26.98 ± 1.97c	90.79 ± 11.25a	99.01 ± 5.55a	76.74 ± 10.64ab
Gly	94.66 ± 4.96a	40.03 ± 4.75c	71.49 ± 4.84b	103.56 ± 4.25a	57.86 ± 5.5bc	95.06 ± 6.72a
β-Ala	27.64 ± 3.83d	51.4 ± 6.95bc	43.67 ± 2.57cd	65.81 ± 2.58b	50.37 ± 4.71bc	89.45 ± 0.79a
Ala	319.32 ± 37.81abc	58.75 ± 16d	307.72 ± 18.44bc	429.43 ± 46.75a	211.62 ± 7.18c	416.52 ± 16.56ab
Pro	89.63 ± 5.96a	46.75 ± 4.55b	56.35 ± 5.16b	104.32 ± 9.16a	56.32 ± 4.93b	108.06 ± 9.85a
Met	17.51 ± 1.62c	43.55 ± 4.87a	20.82 ± 1.33c	39.68 ± 2.23ab	40.77 ± 5.28ab	28.17 ± 1.12bc
Val	398.58 ± 8.98b	243.75 ± 18.72c	407.27 ± 28.43b	395.03 ± 15.7b	362.61 ± 41.12b	510.73 ± 4.5a
Trp	46.75 ± 3.38ab	35.68 ± 9.11bc	32.68 ± 1.05bc	58.82 ± 2.04a	22.92 ± 2.61c	46.85 ± 2.09ab
Phe	81.56 ± 8.13a	70.53 ± 15.59a	47.13 ± 3.65a	82.21 ± 6.88a	67.84 ± 7.51a	81.41 ± 3.92a
Ile	17.38 ± 2.04b	22.69 ± 2.49ab	15.78 ± 1.27b	21.09 ± 0.81b	21.17 ± 2.01b	30.31 ± 1.48a
Leu	20.31 ± 3.06b	33.51 ± 3.04a	19.41 ± 1.78b	30.72 ± 1.43ab	31.9 ± 3.05a	40.98 ± 2.81a

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
