# Peer review of "Multivariate Analysis of Amino Acids and Health Beneficial Properties of Cantaloupe Varieties Grown in Six Locations in the United States"

_plants, 2020, doi:10.3390/plants9091058_

Round 1

Reviewer 1 Report

Comments are enclosed in DOCX file

Reviewer 2 Report

Singh et al. describe changes in the amino acid composition in different Cantaloupe varieties which originate from multiple locations in the US. In addition they are investigating the total phenolics content and the antioxidative potential.

I find the research interesting and it is a very nice example how location can influence plant composition, however there are issues in the manuscript that should be addressed before publication.

First of all, when the authors talk about amino acids in the whole manuscript I assume they mean free amino acids and not also protein bound amino acids (total amino acid content)?

How close is the amino acid composition of melons to the recommended dietary composition for humans, and wouldn’t also amino acids present in proteins influence the nutritional value!

Line 37: While GABA is a neurotransmitter it does not cross the blood-brain-barrier, so if taken up with a food source it most certainly does not act as one in humans. Also is GABA resorbed in the human gut.

Line 50: Melons do not produce glucosinolates, so not a great example to chose here

Line 58: Citrulline and NO are produced from Arginine by NOS, so it is not involved in NO production it is the byproduct. Involved implies for me that it is a precursor of some sort. Also is Citrulline resorbed in the human gut.

Figure 2: In the HPLC chromatogram is the x-Axis time in min, hrs or seconds?

Figure 3: Total or combined free amino acids?

Figure 4: assuming the highest peak at around 10 min is the same in all chromatograms, this peak is actually shifting quite a bit between samples, how sure are you about the identification especially with the cluster of peaks between 22 und 23 min.

Figure 5: How many replicates were analyzed

Line 206: TRP is a precursor to serotonin, but more TRP does not mean more serotonin, so this remark is misleading.

Figure 6: How many replicates?

Line 236/263 and so on: Plants is requesting that SI units being used so convert all imperial units such as °F and inces into °C and cm!

Table 1: How many replicates.

Figure 9: How many replicates?

Figure 10/11: Why is the First plot for Serine up to 400 DPPH and ATBS activity and all others only to 200. Same for Met in CLV with ABTS. Also I would hardly speak of correlation with R values of 0.06 or 0.29 or in general below 0.5. Especially for methionine CLV is completely different from all other varieties, for me this indicates that the antioxidant activity is not correlated with this amino acid, but some other ingredient of the melon pulp.

Figure 12: SAT should be SET

Line 601: 1% formic acid in what Water MeOH, ACN

Reviewer 3 Report

This is a very interesting study about the amino acid profiles and antioxidant activities in melons harvested in different locations of the United States. The paper is very well written and abounds in very interesting and useful results for research in the field.

However, in my opinion, the authors should briefly present other growing conditions (in addition to precipitation and temperature) from each growing locations. I am referring here especially to the soil types and the content in the humic substances from each growing locations, as this could be an important factor in the comparative profiling of amino acids. In fact, there are numerous studies that describe the positive relationships between the content of humic substances of the soil and the quality of agricultural production.

Round 2

Reviewer 1 Report

Authors have answered all the questions and did the asked corrections. The manuscript can be considered not for publication by Plants. 

Reviewer 2 Report

The changes significantly improved the manuscript.